# Ultrasound Imaging-Based Methods for Assessing Biological Maturity during Adolescence and Possible Application in Youth Sport: A Scoping Review

**DOI:** 10.3390/children9121985

**Published:** 2022-12-17

**Authors:** Eva Rüeger, Nicole Hutmacher, Patric Eichelberger, Claus Löcherbach, Silvia Albrecht, Michael Romann

**Affiliations:** 1Department of Elite Sport, Swiss Federal Institute of Sport Magglingen, 2532 Magglingen, Switzerland; 2School of Health Professions, Physiotherapy, Bern University of Applied Science, 3012 Bern, Switzerland; 3Swiss Olympic Medical Center, Swiss Federal Institute of Sport Magglingen, 2532 Magglingen, Switzerland

**Keywords:** ultrasonography, bone age, biological maturity, youth sport, talent development

## Abstract

Bone maturity is an indicator for estimating the biological maturity of an individual. During adolescence, individuals show heterogeneous growth rates, and thus, differences in biological maturity should be considered in talent identification and development. Radiography of the left hand and wrist is considered the gold standard of biological maturity estimation. The use of ultrasound imaging (US) may be advantageous; however, its validity and reliability are under discussion. The aims of this scoping review are (1) to summarize the different methods for estimating biological maturity by US imaging in adolescents, (2) to obtain an overview of the level of validity and reliability of the methods, and (3) to point out the practicability and usefulness of ultrasound imaging in the field of youth sports. The search included articles published up to November 2022. The inclusion criteria stipulated that participants had to fall within the age range of 8 to 23 years and be free of bone disease and fractures in the region of interest. Nine body regions were investigated, while the hand and wrist were most commonly analyzed. US assessment methods were usually based on the estimation of a bone maturity stage, rather than a decimal bone age. Furthermore, 70% of the assessments were evaluated as applicable, 10% expressed restraint about implementation, and 20% were evaluated as not applicable. When tested, inter- and intra-rater reliability was high to excellent. Despite the absence of ionization, low costs, fast assessment, and accessibility, none of the US assessments could be referred to as a gold standard. If further development succeeds, its application has the potential to incorporate biological age into selection processes. This would allow for more equal opportunities in talent selection and thus make talent development fairer and more efficient.

## 1. Introduction

Bone maturity is an indicator that estimates the biological maturity of an individual [1,2] and may differ from chronological age, which is calculated using the current date minus the date of birth. During childhood, but more particularly during puberty, individuals may show very heterogeneous growth rates, and the physiological and psychological changes that occur during the transition from adolescence to adulthood are rapid and pronounced [3,4]. Pediatricians and researchers use bone maturity (maturity stages) or bone age (decimal bone age) estimation to evaluate the growth process for various purposes, for example, defining when treatment can take place, or to estimate age for legal purposes [2,5]. In sports, biological maturity affects physical and cognitive skills. There is evidence that talent selection processes are distorted by differences in biological age [6,7]. Especially in sports where physical components influence performance outcomes, differences in biological maturity must be considered in talent development and identification processes to ensure fairness and equality of chances [7,8,9,10]. In addition, cut-off dates based on chronological age indicate whether an athlete is eligible to compete or enter a category. In the absence of birth certificates or to avoid abuse of the system, bone age estimation can serve as an assessment tool [11,12]. Furthermore, estimates of biological maturity, e.g., age at peak height velocity and predicted adult height, can be estimated, with a certain margin of error, using an equation based on weight, as well as sitting and standing height [13,14,15,16]. Using this information, it is possible to define whether an athlete is an early, normal, or late developer, compared to the average of a specific population. This makes it possible to assess whether certain sports favor the selection and support of athletes at a particular stage of development, or to put systems in place that promote equality and fairness [17].

Doyle et al.’s [18] standards and guidelines provide a broad overview of the existing methods to estimate bone maturity. Currently, the assessment of bone age by the Greulich and Pyle or Tanner and Whitehouse methods using radiography of the left hand and wrist, i.e., the estimation of decimal bone age, are considered the gold standard of imaging techniques. However, even though the radiation dose received during an X-ray is minimal [19], researchers and physicians tend to favor other non-ionizing techniques to avoid the ethical problem posed by radiation. In addition, in many western countries (e.g., Germany and Switzerland) it is a legal requirement to select the method with the lowest radiation intensity from several available methods (Strahlenschutzgesetz (StSG, SR 814.50)). Despite being a non-ionizing technique with validated accuracy, MRI is expensive, time consuming, and less accessible [20]. Therefore, the field of auxology is currently studying sonography, focusing on two different techniques. Ultrasound imaging of bone structure relies on the production of images through high multi-frequency linear transducers that allow one to visualize the composition of growth areas, e.g., the presence of cartilage or ossification centers [21,22]. Imaging of bone anatomy allows the direct visualization of the bony epiphyses and, furthermore, the monitoring of the closing of the growth plate, a crucial diagnostic element for bone maturity estimation. Quantitative ultrasound is another technique by which the properties of bone tissue are analyzed quantitively, for example, by the speed of sound or distance attenuation factor [23,24]. These two sonographic procedures rely on gold standard methods, on existing staging systems, or have been newly developed [25,26]. From a practical, ethical, and economic point of view, ultrasonography seems to present many advantages in various fields of application. However, to date, no ultrasound method has been accepted as the gold standard yet and its clinical utility is still being discussed [20]. Some studies have developed reference values for cartilage thickness in healthy children, mainly to detect juvenile idiopathic arthritis [27,28,29,30,31,32]. However, these measurements were not directly applicable to bone maturity estimation at the publication time.

From the authors’ perspective, there exist several US imaging assessment methods applied for different purposes and in different domains. The age range covered by these assessments differs from one study to the other (birth to adulthood [22]). A common method does not seem to exist, and the need for bone maturity estimation through US in youth sports is still present. As literature that summarizes methods for estimating biological maturity based on ultrasound imaging does not exist to date and there is a need for reasonable, cheap, and practicable methods, the aims of this scoping review were (1) to summarize the existing methods for estimating biological maturity through bone maturity using ultrasound imaging in adolescents, (2) to obtain an overview of the level of validity and reliability of the methods used, and (3) to point out the practicability and usefulness of ultrasound imaging in the field of youth sports.

## 2. Methodology

The literature search for this scoping review relied on the methodological framework of Arksey and O’Malley [33]. The established criteria and six specific steps are (i) identifying the research question, (ii) identifying relevant studies, (iii) study selection, (iv) charting the data, (v) collating, summarizing the data and reporting the results, and optionally (vi) consultation exercise.

### 2.1. Identifying the Research Question

The main research question was “What are ultrasound imaging methods used to estimate bone maturity of adolescents, and more specifically, what body parts are investigated and how are the results analyzed?” Leaving the quality of these studies aside, the second and third questions were: “How valid and reliable are the ultrasound imaging methods?” and “Are there any possibilities to implement these/ultrasound imaging methods in the field of sports?” A wide approach to the context, concept, and population was maintained in order to cover as many articles as possible.

### 2.2. Identifying Relevant Studies and Study Selection

The electronic search was conducted in the PubMed, Mendeley, and Google Scholar databases. After a preliminary search with the search terms ultrasonography, bone age, and puberty, the retrieved studies were analyzed through the Yale MeSH Analyzer to detect appropriate search terms. The following keywords combination was applied in the advanced search function of the electronic databases, using Boolean operators “OR” and “AND”: ((ultraso* OR sonography) AND ((bone OR biological) AND (age OR maturity)) AND (adolesc* OR youth OR puberty)). The search included articles published up to November 2022 and was conducted by two authors (ER and MR). According to the PICOS framework, the first inclusion criterion stipulated that participants had to fall within the age range of 8 to 23 years, which represents the minimal age for the normal onset of puberty [34] and the latest age for epiphysis maturation that occurs at the clavicula [1,35]. The second inclusion criterion was that the participants had to be free of bone fractures or other diseases in the region of interest. Only studies whose objectives were to measure bone maturity using ultrasound were included. Studies involving subjects with diseases affecting skeletal growth were excluded. Quantitative ultrasound was not included. For all selected studies, the titles and abstracts were reviewed first. Secondly, full texts of the potential studies for inclusion were screened. Articles in English, German, French, and Italian were included. Once this initial search was complete, the reference lists were examined to find any new studies that met the inclusion criteria. Finally, the publications of 17 journals dedicated to ultrasonography were examined.

### 2.3. Data Charting

To sort and furthermore analyze the extracted material, a data-charting form was developed using Microsoft Excel. A reviewer (NH) charted the data as follows: Study, year of publication, country, intervention and aim(s), population, domain, methods, examiners, readers and duration of the assessment, and results and conclusion. The articles were separated into two groups depending on whether the ultrasound technique was compared to another accepted technique and method (labelled “validity group”, VG) or whether the ultrasound method itself was tested for its reliability (labelled “reliability group”, RG). Further classification was constructed on the comparison techniques and methods used and the body sites examined. The complete charting form was reexamined by a second author (ER) to ensure the correctness and completeness of the extracted data. The decision for inclusion or exclusion of the studies was then validated by a third author (MR).

## 3. Results

### 3.1. Collating, Summarizing the Data and Reporting the Results

Our first research and analysis process retrieved 53 potentially relevant articles after applying inclusion criteria to the titles and abstracts. These first selected full-text articles were then reviewed. We subsequently excluded 23 articles due to low age span or the use of quantitative ultrasound. One study was excluded as a Master’s thesis containing inappropriate statistics and therefore no reliable results. Two articles were identified as identical, despite different named authors [36,37]. They were included as two independent articles in the analysis, as they were published separately. Finally, 30 articles were included (Figure 1). The data extraction tables (Table 1 and Table 2) report the main content of these articles for the VG and RG, respectively. For the VG, 14 studies were included, seven of which also measured the inter- and intra-rater reliability or agreement for their measures. The RG consisted of 16 studies.

**Table 1 children-09-01985-t001:** Articles retained for the validity group (*n* = 14).

Comparator	US Body Region(s)		Study	Year	Country	Intervention and Aim(s)	Population	Domain	Methods	Examiners, Readers and Duration	Results and Conclusion
X-ray Iliac bone and left wrist	Iliac bone and left wrist	1	Wagner et al. [38]	1995	Germany	Sonographic and radiographic examination of the iliac bone apophysis (Risser’s sign) and the left distal radial epiphysis. Determination of skeletal maturity by ultrasound in order to reduce ionizing radiation to the growing skeleton.	5–19 years of age 49 girls, 15 boys Idiopathic scoliosis	Pediatrics	US of ilium: Risser Grade (0–V) US of left wrist: Radial epiphysis open or closed (yes-no) X-ray of ilium: Risser Grade (0–V) X-ray of left wrist: Greulich and Pyle (atlas)	-	Valid Applicable
X-ray Iliac bone	Iliac bone	2	Thaler et al. [39]	2008	Austria	Determination of the accuracy of ultrasound evaluation of the Risser Grade as compared to plain radiography in patients with adolescent idiopathic scoliosis.	7–17 years of age 36 females, 8 males Idiopathic scoliosis	Pediatrics	US and X-ray of ilium: Risser Grade (0–V)	US and X-ray: senior staff skeletal radiologists	Valid Applicable
X-ray Iliac bone	Iliac bone	3	Torlak et al. [40]	2012	Turkey	Assessment of the efficiency of ultrasonographic evaluation of Risser Sign compared with radiographic evaluation, and investigation of intraexaminer and interexaminer reliability of ultrasonographic evaluation.	10–17 years of age 70 females, 72 males Minor pelvic trauma or scoliosis	Pediatrics	US and X-ray of ilium: Risser Grade (0–V)	US and X-ray: two orthopedists	Valid Reliable Applicable
X-ray Iliac bone	Iliac bone	4	Chauhan et al. [41]	2019	India	Sonographic and radiographic examination of the Risser Grade. Comparison of sonographic and radiographic epiphyseal iliac crest ossification for age estimation in living.	10–22 years of age 28 females, 32 males Healthy	Pediatrics	US and X-ray of ilium: Risser Grade (0–V)	-	Valid Applicable
X-ray of left hand and wrist	Femoral head	5	Castriota-Scanderbeg et al. [42]	1998	Italy	Comparison of sonographically assessed thickness of femoral head cartilage and skeletal age determined by the GP and TW2 left hand radiograph by establishing the level of agreement between methods, the differences between the calculated skeletal age and chronological age, and the sensitivity, specificity, and predictive values of each method.	1.3–21.3 years of age 56 females, 59 males Proven or suspected growth disorder	Pediatrics	US of hip: Femoral head cartilage thickness, skeletal ages derived from normal values obtained in a healthy Italian population (distance) X-ray of left hand and wrist: Greulich and Pyle, Tanner and Whitehouse II (atlas)	US: pediatric radiologist X-ray: experienced pediatric physician	Valid Applicable
X-ray of left hand and wrist	Wrist, knee, ankle	6	Wan et al. [43]	2020	China	Clarification of the correlations between the sonographic ossification ratios of the wrist, knee, and ankle, and the radiographic bone age in patients from infants to teenagers. Development of a new parameter to evaluate bone age with ossification ratios from bones with relatively higher correlations.	0–19 years of age 139 females and 132 males No pathologic modifications of the wrist, knee and ankle	Pediatrics	US of wrist, knee and ankle: Ossification ratio X-ray of left hand and wrist: Tanner and Whitehouse III (atlas)	US examination: operators with experience for 1 and 3 years and trained with the protocol. US evaluation: radiologists 2.6 min	Valid Reliable Applicable
X-ray of left hand and wrist	Left wrist and knee	7	Wan et al. [26]	2021	China	Construction of score-for-age normal values and determination of the diagnostic performances of the method. Evaluation of ultrasonic bone age of the left hand and knee of pathologic patients with normal values of score for age. Comparison with X-ray assessment.	0–19 years of age 511 females, 578 males Normal value group: without clinical diseases potentially affecting skeletal growth Validation group: clinically suspected growth disturbance	Pediatrics	US of left wrist and knee: Ossification ratio and the skeletal maturity score X-ray of left hand and wrist: Tanner and Whitehouse III, Greulich and Pyle (atlas)	US examination and evaluation: radiologists with 20, 6, 5, 1 years of experience and trained with the protocol X-ray: radiologists with 2 and 10 years of experience in bone age radiography evaluation 2 min ± 2	Valid Reliable Applicable
X-ray of left hand and wrist	Left hand and wrist (GP + stages)	8	Ağırman et al. [36]	2018	Turkey	Assessment of the fit between the direct radiography and ultrasonography findings from the left hand–wrist and investigation of whether bone age and pubertal growth excretion are detectable with ultrasonography without ionizing radiation.	10–17 years of age 82 females, 38 males Healthy	Dentistry	US and X-ray of left hand and wrist: Greulich and Pyle (atlas) and scoring system (I–V).	X-ray: technician with at least 5 years of working experience 2–3 min	Valid Reliable Applicable
X-ray of left hand and wrist	Left hand and wrist (GP + stages)	9	Razak and Meena [37]	2018	India	Assessment of the fit between the direct radiography and ultrasonography findings from the left hand–wrist and investigation of whether bone age and pubertal growth excretion are detectable with ultrasonography without ionizing radiation.	10–17 years of age 82 females, 38 males Healthy	Dentistry	US and X-ray of left hand and wrist: Greulich and Pyle (atlas) and scoring system (I–V).	X-ray: technician with at least 5 years of working experience 2–3 min	Valid Reliable Applicable
X-ray of left hand and wrist	Left hand and wrist (SMS and OR)	10	Wan et al. [22]	2019	China	Assessment of the relationship between ultrasonic determination of ossification ratio and standard radiographic bone age from birth to near adulthood. Potential provision of a quantitative modality for estimation of bone age by conventional ultrasound.	0.1–19 years of age 94 females and 78 males No pathologic modification of the hand and wrist	Pediatrics	US of left hand and wrist: Ossification ratio and skeletal maturity score. X-ray of the left hand and wrist: Tanner and Whitehouse III (atlas)	US examination: sonographic imaging specialist US evaluation: radiologists with experience in musculoskeletal ultrasound for 1, 2, and 3 years and trained for the protocol X-ray evaluation: radiologists 4–5 min	Valid Reliable Applicable (with caution)
X-ray of left hand and wrist	Hand and wrist (stages)	11	Nessi et al. [44]	1997	Italy	Examinations of the centers of ossification of the hand and wrist in adolescent by ultrasonographic compared to radiographic evaluation. Determination of the growth phases.	7–16 years of age 26 patients Difference between physical development and chronological age	Dentistry	US and X-ray of the hand and wrist: Fishman stages (0–II)	US and X-ray: radiologists	Not valid Not applicable
X-ray of left hand and wrist	Hand and wrist (stages)	12	Giuca et al. [45]	2002	Italy	Comparison of the results of a sonographic and radiographic evaluation of the left hand and wrist.	9–18 years of age 11 females, 14 males Delayed or precocious skeletal development	Pediatrics	US and X-ray of left hand and wrist: detection of the presence of growth cartilage (yes or no)	-	Not valid Not applicable
CT	Clavicular epiphyses	13	Gonsior et al. [46]	2013	Germany	Comparison of the staging results for both clavicles of the same subjects by sonography, computed tomography, and macroscopy.	15.8–28.8 years of age 5 males Corpses without trauma of the clavicular epiphyses or cranial sternum region nor diseases affecting ossification process	Forensic medicine	US of the clavicular epiphyses: Classification following Schulz et al. (I–IV) CT of the clavicular epiphyses: Classification following Webb and Suchey (I–IV) Autopsy of the clavicular epiphyses: Classification following Webb and Suchey (I–IV)	US: one prepared and experienced examiner	Not valid Not applicable
MRI	Right knee	14	Herrmann et al. [25]	2021	Germany	Test of the feasibility of a US-based method for assessment of epiphyseal growth plate closure around the knee for forensic age estimation and comparison of the findings to MRI.	14.4–19.3 years of age 33 males Healthy	Forensic medicine	US of the knee: Classification by stages (I–III) MRI of the knee: Classification following Jopp et al. (I–III)	US examination: radiologist MRI evaluation: readers with 5 years of experience in forensic medecine 2.65 ± 2.72	Valid Reliable Applicable

**Table 2 children-09-01985-t002:** Articles retained for the reliability group (*n* = 16).

US Body Region(s)		Study	Year	Country	Intervention and Aim(s)	Population	Domain	Methods	Examinators, Readers and Duration	Results and Conclusion
Clavicle	1	Benito et al. [35]	2018	Spain	Determination of the fusion time of both sternal ends of the clavicle by ultrasonography. Evaluation of whether it may be used to estimate the legal age of adulthood in Spain. Reduction of minors’ exposure to radiation.	5–30 years of age 146 females, 75 males	Forensic medicine	Sternal end of both clavicle: classification by Schulz et al. (I–IV)	-	Not reliable Applicable (with caution)
Clavicle	2	Quirmbach et al. [47]	2009	Germany	Assessment of whether the system could be used to evaluate the degree of ossification of the medial clavicular epiphyseal plate (both sides). Establish at what age full ossification could be demonstrated. See if this criterion, as proof that 21 years of age had been reached, could be demonstrated with the necessary degree of reliability required by criminal law.	18–24 years of age 77 males Healthy	Forensic medicine	Both medial clavicular epiphyseal plate: classification by Schulz et al. (I–IV)	Examiners prepared for the experiment and trained for the method	Not reliable Not applicable
Clavicle	3	Schulz et al. [48]	2008	Germany	Determination of whether the ossification stage of the right medial clavicular epiphyses can also be determined by ultrasonography.	12–30 years of age 39 females, 45 males Healthy	Forensic medicine	Right medial clavicular epiphyses: classification by Webb and Suchey (I–IV)	Physician qualified and certified	Reliable Applicable
Clavicle	4	Schulz et al. [49]	2013	Germany	Examination of the time frame of the ossification of right medial clavicular epiphysis in a large number of cases.	10–25 years old 307 females, 309 males Healthy	Forensic medicine	Right medial clavicular epiphysis: classification by Schulz et al. (I–IV)	Qualified arthrosonographist	Reliable Applicable
Clavicle	5	Gonsior et al. [46]	2016	Germany	Evaluation of the stage of ossification of the medial clavicular epiphysis for both sides. Assessment of whether the determination of complete union of the medial clavicular epiphysis could be used as a criterion to prove that an individual had attained the age threshold of 18 years.	14–26 years of age 215 females, 195 males Healthy	Forensic medicine	Both medial clavicular epiphysis: classification by Schulz et al. (I–IV)	Experienced or prepared examiners	Not reliable Not applicable
Humerus	6	Sánchez et al. [50]	2017	Spain	Determination whether the process of ossification of the proximal humeral epiphysis can be observed using the ultrasound technique and whether studying this is of any use in estimating legal age.	5–30 years of age 146 females, 75 males	Forensic medicine	Proximal humeral epiphysis: classification in stages (0–V)	Forensic anthropologists and researcher	Reliable Applicable
Elbow	7	Schulz et al. [51]	2014	Germany	Examination of whether ultrasound examination of the ossification of the right olecranon could be used for the purposes of age estimation.	10–25 years of age 307 females, 309 males Healthy	Forensic medicine	Right olecranon: classification by Schulz et al. (I–IV)	Physician qualified and certified in the area of arthrosonography	Reliable Applicable
Distal radius	8	Ekizoglu et al. [52]	2021	Turkey	Ultrasonographic evaluation of ossification of the left distal radius epiphysis to show its utility in forensic age estimation in living individuals. Assessment of the usability of US, as a nonionizing method, for pediatric age groups. Validation of the methodology of Schmidt et al.(2013) and comparison of the result obtained by those authors to Turkish population.	9–25 years of age 366 females, 322 males Healthy	Forensic medicine	Left distal radius: classification by Schulz et al. (I–IV, modified)	Observers with 10 and 2 years of experience in forensic age estimation	Reliable Applicable
Distal radius	9	Schmidt et al. [53]	2013	Germany	Verify the potential of ultrasound techniques for use in assessing ossification of the right distal radial epiphysis and its chronological dependency as discovered in the course of the pilot study.	10–25 years of age 306 females, 309 males Healthy	Forensic medicine	Right distal radial epiphysis: classification by Schulz et al. (I–IV)	Physicians with experience in imaging procedures in forensic age estimation and certified	Reliable Applicable
Distal radius	10	Karami et al. [54]	2014	Iran	Evaluation of the diagnostic accuracy (with a focus on sensitivity) of the ultrasonography in bone age determination with measuring the thickness of growth plate in the distal radius. Identification of subjects having growth plate width ≤ defined cut-off (positive test) and are actually over the determined age in each category according to the identity documents.	15–20 years of age 82 males Healthy	Sport	Width of distal radial epiphysis, cut-off point for each category (distance)	Radiographist	Reliable Applicable
Distal radius	11	Karami et al. [55]	2016	Iran	Ultrasonographic examination of the epiphysis of the left distal radius. Evaluation of the effectiveness of ultrasound-based methods in a larger and more diverse socioeconomic group of older children, where the accuracy of this method seems to be least.	14–18 years of age 100 females, 100 males Healthy	Sport	Width of left distal radial growth plate (distance)	Radiology residents	Reliable Applicable (with caution)
Iliac crest and olecranon	12	Pitlovic et al. [56]	2013	Croatia	Ultrasonographic examination of the iliac crest and the olecranon apophysis. Test of whether assessment of olecranon apophysis ossification by ultrasound has value in prediction of annal growth and peak height velocity.	10–15 years of age 134 subjects Healthy	Pediatrics	Iliac crest: Risser grade (0–V) In subjects graded as Risser 0, olecranon apophysis: additional classification (0–VI)	Orthopedic surgeon and general surgeon	Not reliable Not applicable
Iliac crest	13	Schmidt et al. [57]	2011	Germany	Pilot-analysis of the forensic applicability of a sonographic evaluation of the apophyseal ossification of the iliac crest for skeletal age assessment.	11–22 years of age 16 females, 23 males Healthy	Forensic medicine	Iliac crest: classification by Schulz et al. (I–IV)	Examiner certified in the field of skeletal sonography	Reliable Applicable
Iliac crest	14	Schmidt et al. [53]	2013	Germany	Examination of the value of skeletal sonography in assessing the age-dependent process of ossification of the apophysis of the Crista iliaca in a more extensive population.	10–25 years of age 307 females, 309 males Healthy	Forensic medicine	Iliac crest: classification by Schulz et al. (I–IV)	Physicians with experience in imaging procedures used in forensic age estimation and certified	Reliable Applicable
Elbow and wrist	15	Shedge et al. [58]	2021	India	Establishment of the applicability of US, a non-invasive and safe technique, to visualize ossification centers of the left wrist and elbow joints for their appearance and fusion among boys between 14 and 17 years of age in the Ahmednagar region of India.	13.73–17.04 years of age 31 males Healthy	Pediatrics	Left wrist and elbow: classification by Schmeling et al. (I–V)	Researcher	Reliable Applicable
Wrist, second metacarpophalangeal joint, knee, ankle	16	Windschall et al. [59]	2020	International	Ultrasonographic examination of the wrist, second metacarpophalangeal joint, knee and ankle vascularization, and their ossification grade. Assessment of the intra- and interobserver reliability of identification of normal joint vascularization in healthy children in different age groups and evaluation of the intra- and interobserver agreement of a new scoring system for assessing the grade of maturation of ossification nuclei in healthy children.	2–16 years of age 5 females, 7 males Healthy	Pediatrics	Wrist, second metacarpophalangeal joint, knee and ankle: classification in stages (0–IV)	Minimum two years of expertise in pediatrics US	Reliable Applicable

### 3.2. Validity, Reliability, and Acceptance

The assessment methods could be classified into four main categories: (i) Assessment of the left hand-wrist compared to images of an atlas (Greulich and Pyle method), (ii) computation of a skeletal maturity score, (iii) staging the ossification process, and (iv) measurement of distance or ratio of measured distances, e.g., of the ossification center, epiphysis, or cartilage thickness. The first category provides an estimate of bone age, the second category derives bone age from a score, and the last two categories provide an estimate of bone maturity through categorization into age categories [60,61].

For the VG, a total of seven body regions were investigated for assessing bone maturity or age: The iliac bone, femoral head, wrist and hand, clavicle, knee, and ankle. The body regions most investigated by both X-ray and US were the hand and wrist. The US imaging assessment method more frequently estimated a stage of bone maturity instead of a bone age (76.7% of the assessments).

In the RG, eight body regions were investigated in different studies to test the reliability of the US measurements. These were the clavicle, wrist, elbow, iliac bone, ankle, hand, knee, and shoulder. The clavicle was the region most commonly investigated (five assessments). The assessment methods were almost all based on classifications by the bone maturity stage, with only two studies using the distance measured at the growth plate of the distal radius to classify the participants into age categories. The results obtained were therefore only estimates of bone maturity and not of bone age.

In the VG, the different US methods were statistically evaluated by the authors and were acceptable in 10 studies (71.4%), to be investigated further in 1 study (7.1%), and not valid or reliable enough in 3 studies (21.4%). The latter studies included methods assessing the femoral head cartilage thickness [42], the maturation of the clavicular epiphysis according to the stages of Schulz et al. [46], and the presence or absence of a growth plate on the distal radius and hand bones [45]. In the first two studies, agreement with the gold standard was too low or the transfer from the staging system of the comparative method could not be transferred to US imaging. In the last study, the use of US imaging was recommended as a complementary method to standard radiography. The seven studies that additionally measured the inter- and intra-rater reliability in the validity group all reached high to excellent reliability between and/or within examiners.

In the RG, the US measurements were considered reliable in 11 studies (68.8%), to be applied with caution in 2 studies (12.5%), and not sufficient in 3 studies (18.8%). Two of the insufficient studies measured the maturation stage of the medial clavicular epiphysis based on the stages of Schulz et al. [48]; however, these were in the forensic medicine domain, in which only the decision of the age threshold is critical [47,62]. The third study measured the maturation stages of the olecranon epiphysis [56]. In this study, the method was not trustworthy due to the small sample size leading to a non-significant difference between stages and growth velocity.

### 3.3. Usability, Practicability, and Economy

The main area in which the studies were conducted was forensic medicine, in which 13 studies (43.3%) aimed to estimate legal age, followed by pediatrics with 12 studies (40%). The main aims of these studies were to uniquely identify bone maturity or obtain information necessary to adjust the treatment of idiopathic scoliosis. Three studies were further conducted in dentistry (10%), where bone age is important for estimating the period and type of treatment. Sport was the least-represented field with only two studies (6.7%) aiming to develop a method to control chronological age reporting and avoid cheating.

Nine studies mentioned the origins of the participants or the composition of the sample. The 21 other studies did not report any information on the origin of the participants, and in this case, it was assumed that they came from the country in which the study was conducted. From this, 19 studies (63.3%) were conducted with European participants (Germany, Italy, Spain, Austria, and Croatia) and 11 studies (36.7%) with Asian (India, China, Iran, and Turkey, as it is mainly part of it) participants. The number of participants strongly differed between studies, ranging from 5 to 1089 participants. In two studies, gender was not mentioned. The average age of the subjects ranged from 9.2 (SD = 4.8) to 21.6 (SD = 4.6) years, covering the entire puberty period.

The duration of measurements, as an important variable for practicability, was mentioned in 40% of the studies in the VG and 0% in the RG mentioned. Thus, on average, the duration was 2.79 min (SD = 0.87).

## 4. Discussion

The first aim of this scoping review was to examine and summarize the different methods used to estimate biological maturity by ultrasound-based imaging in adolescents. In the 30 studies selected for the review, 4 main methods were listed and 9 different body parts were investigated (Table 3), highlighting the diversity in directions taken in search of a valid and reliable method to estimate bone maturity by US imaging. The second aim was to obtain an overview of the level of validity and reliability of the methods used. Despite the promising start of the results in this review (70% of methods considered as applicable), their validity and the choice of the body region investigated are still under discussion. Thus, none of the methods have yet been defined as the method of choice in the estimation of bone maturity or age. The inter- and intra-rater reliability was high in all studies, demonstrating the repeatability of measurements and estimates. The third aim of this scoping review was to discuss to point out the practicability and usefulness of US imaging in the field of youth sports. In this context, it could be shown that four different analytical procedures exist in the literature. In addition, knowledge of the biological age is a crucial component for fair selection and for the implementation of bio-banding in youth sport. Furthermore, existing methods could be found to be too inaccurate (e.g., anthropometric measurements), too expensive (e.g., MRI), or too radiation-intensive (e.g., X-ray). In contrast, ultrasound was described as practicable, cheap, and radiation-free.

### 4.1. Validity, Reliability, and Acceptance

One of the most significant advantages of using ultrasonography is the absence of ionization. According to human research and age estimation procedure legislations, the risks and intrusiveness must be reduced to a strict minimum and the technique used must prioritize a lack of radiation [63,64]. Although adult radiation exposure is minimal in an X-ray of the extremities, i.e., 0.001 mSv compared to 0.27 mSv for one year of terrestrial radiation [19,65], repeated measurements for longitudinal growth monitoring should be avoided. Thus, ultrasonography would be advantageous for biological maturity estimation, as long as the accuracy of the measurements is higher than other non-invasive methods such as anthropometric measurements. MRI is a technique that has been validated but is not generally considered a reference yet, as its usefulness has to be confirmed, and further studies with higher numbers of participants are needed [20,66]. In addition, its high costs and time consumption hinder the implementation in the field of sport, particularly in youth sport.

Of all the methods presented in the studies, 70% were considered to be acceptable, with relatively high validity and reliability. The comparison to the gold standard showed positive perspectives, although in two cases (femoral head thickness and maturation stages of clavicular epiphysis), the agreement was statistically unsatisfactory. To be accepted, the staging systems have to achieve the precision required by the goal of the assessment (e.g., the limit of age or growth monitoring). However, an estimation of decimal bone age by US imaging is lacking. For greater accuracy, the estimation of bone maturity could be combined with additional measurements of other body regions or include anthropometric parameters to prevent unprobeable deviations of results [67].

The US-based skeletal maturity score method developed by Wan et al. [22,26,43] provides interesting results. Measurements at the wrist and knee allowed them to reach values corresponding to the chronological age of healthy subjects. Furthermore, the method was also tested on subjects with growth disorders with valid results compared to the gold standard of hand radiographs and estimation by the Tanner-Whitehouse 3 and Greulich–Pyle methods. Despite the restricted classification of the maturity stages (*n* = 3), the study by Herrmann et al. [25] suggests the development of an atlas using the ultrasound scanning technique of the knee joint. Indeed, the creation of five images per zone (medial distal and lateral distal femoral physis, medial proximal and lateral proximal tibial physis, and lateral proximal fibular physis) provides a fairly complete overview of the zone.

Currently, the accuracy of US measurements depends heavily on the examiner’s expertise and anatomical knowledge. The focus of future research in this area should therefore be the good standardization of the procedure and the objectification of the image analysis. In this sense, the aim must be to improve inter- and intra-rater reliability and simplify the procedure for researchers through good standardization.

### 4.2. Usability, Practicability, and Economy

Several areas of research were identified in the various studies. From a pediatric, legal, or sporting point of view, there is interest in developing a non-ionizing technique to assess biological maturity. Furthermore, orthodontic support, monitoring of idiopathic scoliosis during adolescence, and growth monitoring require repeated measurements, and as such, would profit from a non-ionizing and cheap technique. In this context, it could be shown that four different analytical procedures exist in the literature. In addition, knowledge of the biological age is a crucial component for fair selection and the implementation of bio-banding. Furthermore, existing methods could be found to be too inaccurate (e.g., anthropometric measurements), too expensive (e.g., MRI), or too radiation-intensive (e.g., X-ray). In the field of sports, the organization of systems based on biological age, such as the right to participate in competitions or bio-banding, does not represent a need for medical diagnosis per se, and therefore, it may be more difficult to allow ionizing technologies from a legal point of view. A valid method of estimating biological maturity by ultrasound would thus be a beneficial alternative.

More specifically, in the field of sport [9,12], the distribution of athletes into chronological age classes often creates imbalances between competing adolescents. Thus, the overrepresentation of early maturers and relative age effects are very common, i.e., children born at the beginning of the year are overrepresented in competitive sport compared to those born at the end of the same year [6,68,69]. This effect progressively lessens closer to the end of growth. In addition, at the onset of puberty, a disparity in performance capacity linked to the biological development of the athletes arises. For example, studies show that among soccer national team players under the age of 15, early developers are faster, more powerful, more likely to win duels, and have higher chances of being selected for talent development programs [70,71]. Conversely, late developers selected for superior teams often show superior technical abilities [71,72]. Bio-banding is a form of play in which players are divided into teams according to their biological maturity in order to mitigate differences associated with maturity status and ensure equality [72]. If the estimation of biological maturity by ultrasound proves to be more accurate than anthropometric methods, it could support such systems to ensure fairness between young athletes during competitions and selection. Monitoring growth throughout the puberty period would also help to improve and individualize training and possibly reduce the risk of injuries, especially around peak age velocity [73,74].

Depending on the field in which the method is used, the need for precision in the estimate may differ. In forensic medicine, for example, the method should be most accurate for determining a chronological age representing the majority, which is crucial for law application [63]. In the field of sport, however, growth velocity (tempo) and age at peak height velocity (timing) are the most interesting for defining the biological status of an athlete [75]. The key is to be able to categorize players showing a normal, fast, or slow growth velocity, or to define their developmental stage in order to adapt loads, restrict overloading of growth areas, and thus, possibly reduce the risk of injuries (Morbus Scheuermann, Osgood–Schlatter, Sever’s disease), and organize adjusted competition categories.

According to the results of the review, many studies have developed methods that define a growth stage rather than a precise bone age (76.7% estimates for the VG and 100% in the RG). The number of growth stages ranged from a minimum of 3 to a maximum of 7. If we consider the age range of 8 to 23 years, which is the maximal range in which normal puberty and ossification processes occur, the theoretical maximum number of growth stages would divide the individuals into delimited categories of 2.1 years. In only two studies was the number of stages bigger than the age span of the participants, allowing the authors to reach a precision smaller than one year for bone age estimation [56,58]. Given that classification into biological developmental stages can be performed by anthropometrical measurements within an age range of one year [15] and that biological age can be estimated on a 0.1 year-scale, most ultrasound methods have to be refined to reach at least the same precision. In youth sport, this precision is particularly necessary because differences in performance can already be observed between athletes born 6 months apart [76].

In addition to the great diversity in methods, the aspect of different ethnicities must also be considered when incorporating anatomical variability and growth differences in the estimates [77,78]. The most promising methods should then be tested on different ethnic groups to generalize the results.

Compared to MRIs (approximately 20 min [25]), the duration of the US examination (mean = 2.79 min, SD = 0.87) is advantageous. However, the duration of the measurement and the estimation also depend on the expertise of the examiner. Of the 86.6% of studies that mentioned the level of expertise of the examiners, the measurements in only two studies were conducted by individuals who were not explicitly affiliated with a medical imaging profession or in a domain requiring expertise in estimating biological maturity [47,62]. Each of the four examiners in question had been trained by the DEGUM (German Society for Ultrasound in Medicine) introductory course to the locomotor system. However, the methods tested in these studies were assessed as invalid. The expertise of the reviewers was not questioned, as in both cases, the field of research was forensic medicine, where the definition of a legal age limit requires great precision. The expertise of the examiners therefore must be further investigated, which could, for example, be performed by examining the inter-examiner reliability of different expertise levels. Furthermore, no study reported the use of a handheld device, which would be a significant benefit in the field of sport, to facilitate field implementation and restrict the budget for purchasing such technology.

## 5. Perspectives

The current review demonstrates the vast number of possible methods for estimating bone maturity. This knowledge should be further explored to develop a reliable and valid method, with the aim of achieving gold standard status. For this purpose, standard planes for ultrasounds of the specific bone areas must be clearly defined. This includes the standard positioning of the limb examined during the examination. The addition of investigations on several regions of the body and the combination of different methods could help to improve the accuracy of the estimation.

Furthermore, future investigations regarding the refinement of maturity stages or a method directly measuring bone age to obtain the required precision in youth sport, as well as the inter-rater reliability of the assessment, level of expertise, and measurement accuracy of handheld devices, should be conducted. The field of application and the purpose of the measurement must be clearly defined, as the difference in existing methods can lead to very divergent results that are not applicable in all situations.

Quantitative ultrasound technology is a promising approach to be considered, and a device has been developed to perform measurements on the wrist. This technology has been validated [23,24]; however, the accuracy of the measurement has yet to be improved, as it is no more accurate than anthropometric methods [67]. As the growth plate is a three-dimensional structure and its bone surface is irregular, methods based on sound speed are prone to errors. However, it would be interesting to compare this technology to ultrasound imaging and possibly combine the advantages of both technologies.

## 6. Limitations

Since the aim of scoping reviews is different from that of systematic reviews, an analysis of the quality of the methodology or risk of bias was not conducted [79].

## 7. Conclusions

This is the first review of ultrasound imaging for assessing maturity. While ultrasound imaging of the wrist and the knee show promising results, none of the ultrasound assessments investigated can be referred to as a gold standard yet, as further validation studies are required. The diversity of the methods, body parts investigated, and the goals sought in the various domains of application do not allow the determination of which method could be developed into the gold standard.

Future studies should carefully analyze the sources of bias that may emerge and aim to develop standardized study designs, considering the diversity between ethnicities, gender, the expertise level of the examiners, the measurements of different body regions, and the combination of several methods and/or ultrasound technologies. The development of such a method would be interesting for the field of sport, due to the absence of ionization, its accessibility, its lower costs, and the rapidity of assessment. Its application has the potential to incorporate biological age into selection processes. This would allow for more equal opportunities in talent selection and thus make talent development fairer and more efficient.

## Figures and Tables

**Figure 1 children-09-01985-f001:**
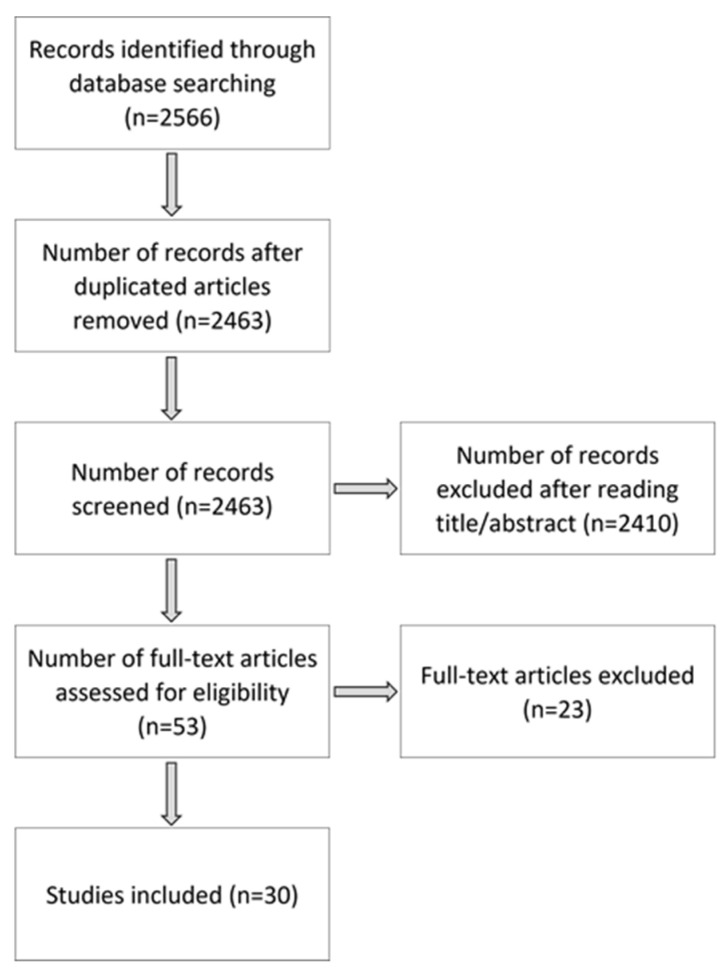
Article selection process.

**Table 3 children-09-01985-t003:** Number of assessments for each method and body region.

Technique		US				RX		MRI	CT	Autopsy	N Assessments/ Body Regions
	Method	Stages	Bone Age (Atlas)	Maturity Score	Distance and Ossification Ratio	Stages	Bone Age	Stages	Stages	Stage	
Body Region											
Ankle		1			1						2
Clavicula		6							1	1	8
Elbow		3									3
Femoral head					1						1
Hand		5	2	1		4	8				20
Iliac bone		7				4					11
Knee		2			2			1			5
Shoulder		1									1
Wrist		9	2	2	5	4	8				30
N assessments		34	4	3	9	12	16	1	1	1	81

## Data Availability

Data are contained within the article.

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
