# Peer review of "Ultrasound Imaging-Based Methods for Assessing Biological Maturity during Adolescence and Possible Application in Youth Sport: A Scoping Review"

_children, 2022, doi:10.3390/children9121985_

Round 1

Reviewer 1 Report

Dear Authors,

I hope you are doing very well.

Congratulations for the work developed so far. The paper is well-written, and the theme sounds pertinent to me! However, from a methodological viewpoint, I believe this work can be largely improved. And by now, this is my main concern. I hope the comments provided can help you dealing with these issues.

All the best

Author Response

Dear Reviewer 1,

We wish to express our gratitude for the positive, constructive and helpful recommendations concerning our manuscript. We appreciate the comments/suggestions, as well as the time and effort that has been expended to enhance the quality of our paper. In particular, the constructive notes on questioning methodology have developed and improved the paper. You and reviewer 2 highlighted the high relevance to a general audience. The review has now been modified in accordance with reviewer recommendations. We have been able to address all queries/points raised. Alongside the manuscript, please find a point by point response to the comments and queries raised. Modifications in the manuscript are highlighted with track changes.

#1: Thank you for this helpful comment. We agree and adapted the sentences accordingly. (line 44).

#2: Yes, we agree. This goal is too ambitious. We have now adjusted the objective in line with your comments and those of Reviewer 2. (line 96).

#3: We added information about the practical and legal need. In addition, we have tried to show the additional scientific benefit in the introduction.

#4: In pilot searches, the databases (e.g. EBSCO , Sciencedirect) were also searched, but this did not add any value (number of hits was not increased). Therefore, the search in this study was only carried out with the "main" search engines. Very specifically, a search of the EBSCO and SCOPUS databases did not return any additional results. (line 125)

#5: There were no differneces between the searches in 2021 and 2022.  There were no new results that met the inclusion criteria. Now we adapted the section in the methodology mentioning the final and overall literature search.

#6: We added an explanation for the age range and added relevant references. The age range should represents the minimal age for normal onset of puberty and the latest age for clavicular epiphysis maturation, the last bone to reach complete ossification. However, there is controversy in the literature about the exact range, which would distract the reader from the focus of this article. Therefore, we have included references in the article that support the chosen range. (line 127-129)

#7: Yes, we agree and clarified the sentences.

#8: Revised accordingly. (line 132)

#9: This is an invited scoping review. Scoping reviews don't typically include a risk of bias assessment. A key difference between scoping reviews and systematic reviews is that the former are generally conducted to provide an overview of the existing evidence regardless of methodological quality or risk of bias. For this reason, an analysis of risk of bias has been omitted from this article; for a future systematic review, an analysis would be exciting and interesting. For this reason, risk of bias has been omitted from this article; for a future systematic review, an analysis of risk of bias would be exciting and interesting.

#10: Numerical summary is generally part of the results section. The point for “contradictory” result was linked to a master thesis that was found in our literature search and had some inappropriate statistic and no reliable results, therefore we had to exclude that papers.

#11: 17 represented the number of studies with high agreement level for reliability. We deleted this number in the abstract to avoid misunderstanding, as it is not important in this section.

#12: ref added

#13: structure modified

#14: we changed the results from mean and SD to age span

#15: we wanted this information for practicability assessment between the different techniques (US, RX, MRI) or methods.

#16: we clarified the sentence

#17: MRI was validated but its clinical usefulness has to be accepted, and large participation studies have to be conducted to reach same acceptance as RX technique and linked assessments.

#18: adapted in text

#19: introduction modified to create this link.

#20: ref added

#21: we did not want to say that growth causes these injuries, but that during growth, overloads may more frequently cause such chronic injuries

#22: explained for comment #6

#23: adapted in text

#24: ok

#25: removed to perspectives

#26: discussed in comment #4

#27: modified in text

#28: we reworded that part to be more explicit.

Reviewer 2 Report

This is a well-written study discussing an important topic. Biological maturity is a major issue in youth sport, talent identification and development. When working with young athletes, there is an increasing need to assess their maturity to ensure a safer and more optimal training and competition loading according to the individual biological stage. The current manuscript may offer a valuable contribution to this field. A few minor comments to address:

In the introduction I recommend including few words about the main principles of ultrasound methodology to improve understanding for a reader not so familiar with this technology. Also, the terms ultrasound imaging and quantitative ultrasound are not adequately explained. What are their differences in terms of measuring methodology and of their results output? The terms bone maturity and bone age should be also defined.

Based on the included papers a variety of age-range was used in previous research. However, there is no reference to the validity and/or reliability of the measurements according to the examined age-range which is a major aspect in such studies. I think this part deserves more attention and discussion. Based on the existing evidence is there an optimal age-range, which we can recommend to practitioners, for using ultrasound imaging? For example, regarding quantitative ultrasound a validity study has suggested estimation of bone age to be valid within 8.5-16.0 years for boys and 7.5-16.0 years for girls. Please refer to:

Utczas K, Muzsnai A, Cameron N, Zsakai A, Bodzsar EB. A comparison of skeletal maturity assessed by radiological and ultrasonic methods. Am J Hum Biol. 2017 Jul 8;29(4). doi: 10.1002/ajhb.22966. Epub 2017 Jan 17. PMID: 28094893.

According to the authors (Ln 292-297) growth stages may be divided into 6 categories, which is not sufficient for such a wide age-range (8 to 23 years). Therefore, does this mean that this technology is not applicable in sports? Considering that in many cases differences of even less than 0.5 year in biological maturity may represent significant differences in athletic performance. This aspect seems to be slightly underdiscussed lacking clear recommendations regarding the usefulness of this method in youth sport.

Author Response

Dear Reviewer 2,

We wish to express our gratitude for the constructive and helpful recommendations concerning our manuscript. We appreciate the comments/suggestions, as well as the time and effort that has been expended to enhance the quality of our paper. You and reviewer 2 highlighted the high relevance to a general audience the field and that is an original, important, and interesting study. The paper has now been modified in accordance with reviewer recommendations. We have been able to address all queries/points raised. Alongside the manuscript, please find a text adaptation to your comments. We added more information about the technique of ultrasonography and to the definition bone maturity and bone age. Modifications in the manuscript are highlighted with track changes.

This is a well-written study discussing an important topic. Biological maturity is a major issue in youth sport, talent identification and development. When working with young athletes, there is an increasing need to assess their maturity to ensure a safer and more optimal training and competition loading according to the individual biological stage. The current manuscript may offer a valuable contribution to this field. A few minor comments to address:

>> thank for your positive and helpful feedbacks. We really appreciate this kind of review process.

In the introduction I recommend including few words about the main principles of ultrasound methodology to improve understanding for a reader not so familiar with this technology. Also, the terms ultrasound imaging and quantitative ultrasound are not adequately explained. What are their differences in terms of measuring methodology and of their results output? The terms bone maturity and bone age should be also defined.

>>yes, we agree and adapted the article accordingly.

Based on the included papers a variety of age-range was used in previous research. However, there is no reference to the validity and/or reliability of the measurements according to the examined age-range which is a major aspect in such studies. I think this part deserves more attention and discussion. Based on the existing evidence is there an optimal age-range, which we can recommend to practitioners, for using ultrasound imaging? For example, regarding quantitative ultrasound a validity study has suggested estimation of bone age to be valid within 8.5-16.0 years for boys and 7.5-16.0 years for girls. Please refer to:

Utczas K, Muzsnai A, Cameron N, Zsakai A, Bodzsar EB. A comparison of skeletal maturity assessed by radiological and ultrasonic methods. Am J Hum Biol. 2017 Jul 8;29(4). doi: 10.1002/ajhb.22966. Epub 2017 Jan 17. PMID: 28094893.

>> This is an interesting and difficult question. I our review we found several methods applied to different body regions. The large variability and the small numbers of subjects do not allow a serious answer to this question. Further studies in the age ranges mentioned and with significantly higher numbers of subjects must be conducted here. Validity is certainly lowest at the end of the ossification process. During the phase of the growth spurt, however, no clear statement can be made.

According to the authors (Ln 292-297) growth stages may be divided into 6 categories, which is not sufficient for such a wide age-range (8 to 23 years). Therefore, does this mean that this technology is not applicable in sports? Considering that in many cases differences of even less than 0.5 year in biological maturity may represent significant differences in athletic performance. This aspect seems to be slightly underdiscussed lacking clear recommendations regarding the usefulness of this method in youth sport.

>> yes, we agree. This a major issue. At present, the scale of the individual stages is still too large for some measurement methods. This point was taken up in the discussion. Nevertheless, modern ultrasound devices can produce better and better images, so that finer and finer gradations are expected in the future. However, it is clear that this is still a limitation of ultrasound.

Round 2

Reviewer 1 Report

Please, see the document attached

Author Response

Dear Reviewer,

We wish to express our gratitude for the positive, constructive and helpful recommendations concerning our manuscript. We appreciate the comments/suggestions, as well as the time and effort that has been expended to enhance the quality of our paper.

Modifications in the manuscript are highlighted with track changes.

#1: Thank you for this comment. We adapted the abstract and the methods accordingly. (line 18 and line 113).

#2: Yes, we agree. We changed the sentences accordingly. (line 306 and 315).

#3: Changed (line 328).

#4: Changed (line 372-373)

Kind regards and all the best!

Kind regards and all the best!